# Distribution Patterns of Gymnosperm Species along Elevations on the Qinghai–Tibet Plateau: Effects of Climatic Seasonality, Energy–Water, and Physical Tolerance Variables

**DOI:** 10.3390/plants12234066

**Published:** 2023-12-04

**Authors:** Muhammad Umair, Xiaofei Hu, Qi Cheng, Shahzad Ali, Jian Ni

**Affiliations:** 1College of Life Sciences, Zhejiang Normal University, Jinhua 321004, China; xiaofie@zjnu.edu.cn (X.H.); cheercq@zjnu.edu.cn (Q.C.); shahzadali320@aup.edu.pk (S.A.); 2School of Agriculture and Biology, Shanghai Jiao Tong University, Shanghai 200240, China

**Keywords:** gymnosperms, conservation, elevation, species richness, climatic seasonality, energy–water, physical tolerance

## Abstract

Climate change is one of the most prominent factors influencing the spatial distribution of plants in China, including gymnosperms. Climatic factors influence gymnosperm distribution along elevational gradients on the Qinghai–Xizang (Tibet) Plateau (QTP), and understanding how species adapt to these factors is important for identifying the impacts of global climate change. For the first time, we examined the county-level distribution of gymnosperm species on QTP using data from field surveys, published works, monographs, and internet sources. We used simulated distribution data of gymnosperms (N = 79) along the elevational gradients to investigate the overall impact of environmental variables in explaining the richness pattern of gymnosperms. Eighteen environmental variables were classified into three key variable sets (climatic seasonality, energy–water, and physical tolerance). We employed principal component analysis and generalized linear models to assess the impact of climatic variables on the gymnosperm’s richness pattern. Gymnosperm species are unevenly distributed across the plateau and decline gradually from the southeast to the northwest. The altitudinal gradients have a unimodal relationship with the richness of gymnosperms, with the maximum species richness at an elevation of 3200 m. The joint effects of physical tolerance and energy–water predictors have explained the highest diversity of gymnosperms at mid-elevation. Because the richness peak correlates significantly with the wettest month’s precipitation and moisture index, this confirms the significance of moisture on gymnosperm distributions due to increased precipitation during the wet season. Furthermore, our results provide evidence that climatic seasonality factors are involved in the decline of gymnosperm richness at high elevations. A total of 37% of gymnosperm species on QTP are listed as vulnerable, nearly threatened, or endangered, with elevations ranging from 600 m to 5300 m. As a result, we conclude that gymnosperms are at high risk of extinction because of the current climate fluctuations caused by global climate change. Our research offers fundamental data for the study and protection of gymnosperm species along the steepest elevation gradients.

## 1. Introduction

Because of its high elevation and complex terrain, the Qinghai–Tibet plateau is the most sensitive plateau in the world to global climate change [1]. Climate change is having a substantial impact on plateau land cover variation, and alpine habitats are among the most susceptible ecosystems [2]. The Qinghai–Tibet Plateau, the world’s third roof, has become wetter and warmer in recent decades [3], and more than 2 °C warming has been identified as an alarming factor capable of causing irreversible alterations in alpine ecosystems [4]. The most substantial climate changes occurred in the northern QTP [5], and were linked with rainfall variations and snow cover extent reduction, resulting in permafrost deterioration [6]. Studying the pattern of species richness under climate change is crucial to understanding the distribution of organisms and their underlying processes. Because of the QTP’s fragile ecology and sensitivity to global climate change, the dynamics of the plateau’s ecosystems can predict the impact of environmental variables on species richness patterns more accurately and predictably than other places, making it a great location for climate change research [7].

Several theories on species distribution and richness patterns, as well as on their underlying mechanisms, have been proposed in ecological research over the years [8,9]. Examining the species distribution and richness patterns along the elevation gradients, as well as the climate–species richness relationship, are important conservation endeavors [10]. Several studies have been carried out in recent years to examine the linkages between species richness and altitude gradients in order to better comprehend the characteristics of these elevation gradients and to develop effective conservation approaches for plant protection and distribution in the context of climate change [11,12]. According to Rahbek [13], the three main forms of species richness patterns along altitudinal gradients are a mid-elevation richness pattern [14,15,16,17,18], a linear increase in species abundance with elevation [19,20,21,22,23], and a gradual decrease in species abundance with elevation [24,25]. Mid-elevation richness patterns predominate in most studies undertaken in mountainous areas across the world, and they are substantially more prevalent than a monotonically declining trend. These patterns of species richness have been thoroughly investigated in a variety of plants, such as pteridophytes [16], wetland angiosperms [26], and seeded plants [14], along elevation gradients on the QTP, China. Gymnosperms are classified as seeded plants, but their morphology, phenology, habitat, and diversity patterns differ from angiosperms [27]. Gymnosperms are a diverse group of plants with the common characteristic of naked or unprotected seeds and grow in a variety of habitats worldwide [28]. Ecologists have made an effort to define the distribution patterns of species along the steepest elevation gradient on QTP, but a comprehensive knowledge of the mechanisms governing gymnosperm diversity has to be developed.

The substantial association between species richness and climatic variables is an essential concept in identifying richness patterns and measuring the underlying processes of species distribution [16]. Elevational diversity gradients (EDG) have been evaluated using the range of species distribution in response to environmental factors along altitude gradients [11,29]. Environmental parameters such as humidity, temperature, precipitation, and available sunshine were considered crucial determinants in the distribution of plants in mountainous areas [16,30,31,32,33]. Climate change is regarded as the most important element impacting the distribution and richness patterns of organisms along altitude gradients, as climatic factors have a high association with elevation [34]. The diversity of species was higher in regions where climatic changes were induced by terrain and elevation [35]. To assess the impact of global temperature change on ecological diversity, it is essential to understand the processes that cause variations in species distribution along elevational gradients. Ecological researchers have established numerous strategies for identifying the richness patterns of organisms along altitude gradients and the environmental variables that influence it [17].

Multiple hypotheses must be addressed in order to explain species richness patterns because no one factor can fully account for distribution patterns within and between taxa [13]. As a result, we investigated how climatic seasonality (CS), physical tolerance (PT), and energy–water (EW) dynamics influenced gymnosperm richness patterns along elevational gradients on QTP. The “energy–water” concept, which is based on the capacity of plants to absorb both energy and water, is the most recognized and much-argued concept for understanding species richness patterns. The relationship between biological diversity and energy–water (EW) variables has been examined in earlier research [33,36]. Understanding the gymnosperm’s richness patterns along altitudinal gradients necessitates a thorough grasp of the “energy–water hypothesis”. The greater the accessibility of energy and water in an area, the greater the diversity of plant species [37]. At lower elevations, water availability has a significant impact on species, but at higher elevations, the variety of species is favored by the availability of adequate energy [38]. Only a few organisms can tolerate severe stress situations such as aridity and extreme cold, according to the physical tolerance (PT) concept [39]. Tropical plants are predicted to endure extreme stress as a result of harsh climatic periods [17]. Several relevant investigations have also described the richness patterns of different organisms, which supports the physical tolerance theory [40,41,42]. It is widely accepted that plants cannot withstand temperature extremes (cold or hot), nor can they withstand aridity. The “climatic seasonality” concept proposes that species richness is sustained by stable environmental conditions. Seasonal variation in climate is supposed to promote species co-existence by providing additional opportunities for niche formation [43,44].

Geographical and climatic variations have undoubtedly made this region a great location for investigating the richness patterns of species, including gymnosperms, and providing a data set for evaluating the driving variables that can impact species distribution. A variety of evolutionary processes and environmental conditions influence the formation of populations from a regional species pool [45,46]. These features operate as environmental filters, selecting which species can resist the rigors of climatic fluctuation and which are best positioned to optimize dispersal, development, and reproductive opportunities [45,46,47]. Gymnosperms, especially conifers (crucial components of tree species in the forest), are critically endangered and especially vulnerable to climate change [48]. Furthermore, climate change causes the extinction of around 41% of gymnosperm species in China, and the loss of habitat regions under future temperature conditions will result in a shift in coniferous forest structure and the loss of ecosystem services associated with the species. Therefore, we investigate the relative effects of climatic conditions on how gymnosperms are distributed along an altitudinal gradient on QTP. The main objectives of this research are to (i) determine the gymnosperm’s richness pattern along the steepest elevation gradients and (ii) gain insight into the determinant role of climatic seasonality (CS), physical tolerance (PT), and energy–water (EW) dynamics in determining the gymnosperm’s richness along the steepest elevation gradients.

## 2. Results

### 2.1. Regional Differences in the Distribution and Richness Patterns of Gymnosperms

There are 79 species overall, and 7 families and 14 genera (Table 1). The Pinaceae family has the most species (57%), followed by Cupressaceae (23%), Ephedraceae (12%), and Taxaceae (5%), with other families (such as Ginkgoaceae, Gnetaceae, and Podocarpaceae) contributing only 1% (Figure 1a). Likewise, the genus *Picea* has the most species (27%), followed by *Juniperus* (27%), *Abies* (27%), *Ephedra*, and *Larix* (27% each), and *Amentotaxus* and *Cupressus* (27% each), while other genera contribute only 1% (Figure 1b).

According to the regional distribution of gymnosperm species on the QTP, Nyingchi city has the most species-rich area (SR = 42). The three most prevalent species in the studied region are *Ephedra gerardiana*, *Ephedra intermedia*, and *Ephedra monosperma*. Table 1 shows the distribution of *Ephedra gerardiana* in a number of sites with elevations between 3400 and 5300 m, including Qamdo, Xining, Shigatse, Lhoka, Yushu TAP, Haixi Mongolian TAP, Haibei TAP, Lhasa, Nagqu, and Ngari cities. Similarly, *Ephedra intermedia* is found in Nyingchi, Haidong, Qamdo, Xining, Lhoka, Huangnan TAP, Yushu TAP, Hainan TAP, Haixi Mongolian TAP, Lhasa, and Ngari cities, with elevations ranging from 1650 to 4300 m (Table 1, Appendix A). *Ephedra monosperma* is found in the cities of Nyingchi, Haidong, Qamdo, Guoluo TAP, Yushu TAP, Hainan TAP, and Haixi Mongolian TAP, Lhasa, Nagqu, and Ngari at elevations ranging from 3100 to 4900 m. *Juniperus communis* var. *saxatilis* and *J. sabina* are widely spread over the Eurasian continent (Table 1).

According to the International Union of Conservation of Nature (IUCN), 61% of gymnosperms are least concerned (LC), 37% are near threatened (NT), vulnerable, and endangered, and 2% are not evaluated (NE) and data deficient (DD), as shown in Appendix A. The following gymnosperms have been designated as endangered by the IUCN: *Ginkgo biloba*, *Picea likiangensis var. hirtella*, *Amentotaxus assamica*, and *Taxus wallichiana. Abies spectabilis*, *Amentotaxus argotaenia*, *Juniperus communis var. saxatilis*, *J. pingii var. wilsonii*, *J. squamata*, *Larix himalaica*, *L. olgensis*, *L. speciosa*, *Picea purpurea*, *Pinus gerardiana*, *P. wallichiana*, and *Platycladus orientalis* are listed as nearly threatened by the IUCN (Table 1).

### 2.2. Changes in Environmental Variables along the Elevational Gradients

Environmental variables vary along the elevation gradients, as shown in Appendix A. Except for SS% and DI, all physical tolerance (PT) and energy–water (EW) variables exhibited a monotonic decline along the elevation gradients (Appendix A), but climatic seasonality (CS) variables showed a monotonic increase (Appendix A). Pearson correlation coefficient data revealed that, with the exception of DI and PWM, all variables had a substantial positive and negative connection with elevation gradients (Appendix A).

Ternary plots were used to visualize the relationships among three predictor variables, i.e., climatic seasonality (CS), energy–water (EW), and physical tolerance (PT), along the elevational gradients. It is clearly illustrated that physical tolerance predictors were significantly higher at lower elevation gradients (Figure 2a), while energy, water and climatic seasonality predictors were found to be higher at higher elevation gradients (Figure 2a).

The first two components have a 99.6% variance, according to PCA (principal component analysis), whereas the PC1 and PC2 axes have 98.2% and 1.4% variations, respectively (Figure 3a). At lower elevation gradients, a positive correlation was observed between the PC2 component and the physical tolerance (PT) variable (Figure 3b). At higher elevation gradients, a negative correlation was observed between the energy–water (EW) and climatic seasonality (CS) variables (Figure 3b).

### 2.3. Gymnosperm–Climate Relationships along the Altitudinal Gradients

The gymnosperm species richness pattern was undoubtedly unimodal, exhibiting a mid-altitude distribution pattern (*r*^2^ = 0.929) (Figure 4a). The maximum observed number of species (S = 44) was 3200 m. Ternary plots revealed that the energy–water (EW) and species richness associations on the Qinghai–Tibet plateau were significantly stronger (Figure 4b) than the physical tolerance (PT) and climatic seasonality (CS).

Based on the mid-elevation curve at 3200 m, the whole gradient was divided into two groups, 100 m–3200 m (lower elevation gradient) and 3300 m–5300 m (upper elevation gradient), and both were analyzed simultaneously in the principal component analysis (Figure 5). The PCA analysis yielded a total inertia of 18.2 based on the sum of all Eigen values (Appendix A). The first eigenvalue had a high value of 15.43, demonstrating a substantial gradient in determining how environmental factors affect gymnosperm species along the elevation gradient. The first two components of PCA revealed 96.9% variance (PC 1-axis: 81.2%; PC 2-axis: 14.5%), as shown in Figure 5. The PC1-axis (explaining 81.2% of the variation), which mainly included GDD_5_ (*r* = 0.24512), GDD_0_ (*r* = 0.24797), MAT (*r* = 0.23895), MI (*r* = 0.22042), GP (*r* = 0.2395), MAP (*r* = 0.25227), T_max_ (*r* = 0.22566), T_min_ (*r* = 0.25041), MTCO (*r* = 0.24572), MTWA (*r* = 0.2292), PDM (*r* = 0.2317), and PWM (*r* = 0.17458), was positively correlated with SR, while DI (*r* = −0.18591), SS% (*r* = −0.25033), TAR (*r* = −0.55473), MDT (*r* = −0.55473), precipitation seasonality (*r* = −0.55473), and temperature seasonality (*r* = −0.55473) showed a negative correlation with SR (Appendix A).

Pearson correlation coefficients showed that MI (*r* = 0.48) and PWM (*r* = 0.69) showed a significant positive correlation with SR (*p* < 0.05), while GDD_5_ (*r* = −0.32), DI (*r* = −0.55), TAR (*r* = −0.50), MDT (*r* = −0.34), and temperature seasonality (*r* = −0.48) had a significant negative correlation (*p* < 0.05), as shown in Figure 6. We used spatial autocorrelation to investigate the significant link between species richness, elevational gradients (independent variables), and predictor variables (dependent variables) to determine the overall effect of climate seasonality (CS), energy–water (EW), and physical tolerance (PT) variables on species richness along the elevational gradients. Species richness had a strong relationship with energy–water (MI and DI) and physical tolerance (PWM) variables along the elevational gradients (Figure 7a,b). Similarly, temperature seasonality showed a close association with SR (Figure 7c).

A log-linear relationship exists between the surrogate variables (GDD5, MI, DI, PWM, TAR, MDT, and temperature seasonality) and gymnosperm richness (Appendix A), because the richness of gymnosperms over elevation gradients is significantly correlated with GDD_5_, MI, DI, PWM, TAR, MDT, and temperature seasonality factors (Figure 8). Meanwhile, other climate variables (e.g., GDD_0_, MAT, GP, MAP, SS%, Tmax, Tmin, MTCO, MTWA, PDM, and precipitation seasonality) had no significant correlation with species richness (Appendix A). PWM has a strong correlation with gymnosperm richness (R^2^ = 0.47) because the maximum PWM values for gymnosperm species richness range between 132.9 and 135.1 at 3200 m (Appendix A).

## 3. Discussion

### 3.1. Distribution Patterns of Gymnosperms on QTP

The Qinghai–Tibet Plateau (QTP) is a global hotspot for gymnosperm species distribution, accounting for approximately 36.4% and 7.7% of all species in China and worldwide, respectively. Gymnosperm species are unevenly distributed across the plateau and decline gradually from southeast (SE) to northwest (NW) (Figure 9a). Our findings are congruent with the findings of Qi et al. [27], who reported that gymnosperm seed mass decreases from the southeast (SE) to the northwest (NW) regions of China.

Nyingchi city is located in the southeast section of the plateau (Figure 9a), and has an average height of 3100 m, a greater diversity of gymnosperm species, and a vegetation cover of approximately 46.09% [49]. Wetlands and forests are among the city’s natural resources, making it an ecologically biodiverse region [50]. The Hengduan mountain system, located in the southeast (SE) of the plateau and forming a corrugated landscape of steep dividing peaks and deep river gorges, is currently regarded as the planet’s most biologically diverse area [14]. The Hengduan mountainous regions have a higher gymnosperm richness due to their diverse habitats, topographies, and hydrothermal conditions [18]. Spruce forest occurs in the subalpine zone of the Hengduan mountainous regions, at the highest elevation of 3200 on the Qinghai–Tibet Plateau (Appendix A). In southwest America, *Picea-Abies* (spruce-fir) forest grows in the subalpine zone at the highest elevation, accounting for approximately 1.1% of the region’s total area [51].

More than 80% of the plateau’s territory is higher than 4000 m, making it the “third pole of the world”, and the temperature is significantly lower than in other locations at the same height [52,53]. The southern and southeastern QTP have a suitable environment and good climate for plant growth and reproduction due to the South Asian summer monsoon, which transports moisture due to higher precipitation [54,55]. The QTP experiences a concentration of precipitation, with amounts ranging from 1000 mm (in the southeast) to 50 mm (in the northwest) between June and September [56]. The increase in precipitation was most obvious in counties bordering the Tibetan Plateau, such as Zayu, Kangmar, Nagarze, and Lhozhag, where it was 20 mm 10a^−1^ [57]. The distribution of the most diverse gymnosperms in the southern QTP may be connected to climatic conditions and elevation variations.

### 3.2. Effect of Environmental Variables on the Richness Patterns of Gymnosperm Species on the QTP

Gymnosperms are found in China at elevations ranging from 0 to 5300 m [29]. The high richness of gymnosperm species on QTP was recorded at 3200 m, establishing a mid-elevation curve (Figure 4a). Our results are consistent with earlier research conducted in the neighboring Himalayan regions, particularly in China and Nepal [15,17,29]. Gymnosperms’ richness patterns showed a unimodal trend in relation to elevation gradients. Such patterns were ubiquitous in the world’s mountainous regions, including in China [29]. Previous research has found a similar unimodal trend in species richness along the altitudinal gradients in other taxonomic categories all around the world [16,19,20,22,24,31,32,33,38,58,59,60,61,62]. All of our findings confirm previous research and demonstrate that environmental factors influence the distribution of gymnosperm richness over elevation gradients [33,63]; however, this is the first complete study that addresses the impact of climate changes on the richness patterns of gymnosperms and their distribution along elevation gradients on the QTP. The QTP sub-alpine zone encompasses an elevation range of 3100 m to 4000 m above sea level [64] and is defined by moderate and suitable environmental conditions for gymnosperm distribution [29]. To conserve gymnosperm species, it is critical to understand the process that governs the richness pattern along the altitudinal gradients. The current study focuses on the significance of environmental factors in determining the gymnosperm richness patterns along the altitudinal gradients on the Qinghai–Tibet Plateau. The species diversity and distribution along elevational gradients serves as a baseline for assessing population range shifts for conservation.

According to our findings, the richness patterns of gymnosperms on QTP are regulated by energy–water (EW) dynamics. The EW variables (GDD5, MI, and DI) have a log-linear relationship with gymnosperm richness (Figure 6). Pandey et al. [18] discovered that the best predictor for determining the gymnosperm richness pattern in China was energy–water (EW). Similar results were also observed in research undertaken in Himalayan areas to explain gymnosperm richness trends [15,17]. Therefore, the mid-elevation curve of gymnosperm richness may be better described by energy–water (EW) variables (Figure 4), indicating the greater number of species in the mid-altitudes with enough energy and water. In plants, available moisture and energy constantly promote photosynthesis, which affects all physiological processes and broadens the diversity of species [39,62]. According to Kluge et al. [61], the richness of plant species is declining both above and below the ideal temperature. Pandey et al. [18] claim that the Qinling-Daba and Hengduan Mountains, which are abundant in both accessible water and energy, have the highest gymnosperm species richness. The gymnosperm species richness was found to be concentrated to the southeast QTP, which has a suitable climate with an abundance of water and energy (Figure 9c,d).

Gymnosperm richness patterns exhibited a similar trend with MI along elevation gradients and gradually decreased from the southeast (SE) to the northwest (NW) of the Tibetan plateau (Figure 9d). Moisture-related factors are the strongest indicators of plant distribution from sub-alpine to tropical environments, assuming that energy is readily accessible [36]. Moisture availability influenced species richness patterns and the distribution of species in the mid-elevation zone [16,33,65,66], including gymnosperm species [15,17,18,29]. For instance, some species of the Gesneriaceae family in China require high moisture, whereas others are drought-tolerant [67]. A moist environment may support the greatest species diversity by hastening plant growth [36,68].

Gymnosperm richness patterns, on the other hand, displayed an inverse trend with DI index along the elevation gradients (Figure 8), gradually increasing from the southeast (SE) to the northwest (NW) of the plateau (Figure 9e). According to Wang et al. [69], the QTP’s aridity steadily increased from southeast (SE) to northwest (NW) during the last century. Gymnosperms’ ability to acquire water in the Himalayas can be severely limited by the steep slopes and thin soil layers at higher elevations [15]. In *Picea* seedlings, cold air and ice blasting can break leaf cuticles and cause drought stress [70] because cold air and low soil temperatures decrease the root system’s ability to absorb water, which leads to drought stress [71]. According to Fei et al. [72]’s research, gymnosperms are less competitive in the drier region of the eastern US because they have lower growth rates than angiosperms [73]. The majority of gymnosperm species are outcompeted by other groups in harsh climatic conditions; gymnosperms may be constrained to mid-elevations where an ideal temperature and precipitation range may result in enhanced energy and moisture availability and, as a consequence, higher gymnosperm diversity.

The richness pattern of gymnosperms on QTP is best explained by the interaction of physical tolerance (PT) and energy–water (EW) variables. Pandey et al. [17], similarly, found that the combined influence of physical tolerance (PT) and energy–water (EW) variables best explained the richness pattern of gymnosperm species in the Himalayan region of Nepal. The QTP climate is distinguished by a dry season from January to April during the winter and a wet season from June to September during the summer. During the warmest/rainy season, abundant energy and high moisture availability may favor high species richness. According to Qi et al. [27], the wet season in summer is frequently correlated with the tree development and growth season in China and has a stronger impact on the reproduction and growth of trees than winter precipitation. Similar to our findings, they noticed a substantial positive correlation between plant seed mass and the warmest month’s precipitation (PWM), which suggests that with an increase in warm-month precipitation due to variations in climate, the gymnosperm richness may increase at the mid-elevation zone with adequate energy and water. Our research identified the highest peak of PWM at the same elevation of 3200 m (Appendix A), which confirms the maximum gymnosperm abundance at mid-altitudes. Because gymnosperm richness over elevation gradients has a similar tendency, it peaks at 3200 m and gradually declines as gradients decrease or elevate (Figure 4). Thus, the substantial reliance of gymnosperm species on physical tolerance (PT) and energy–water (EW) variables may be due to the dry and wet circumstances prevalent in the Himalayan area [74].

Climate seasonality (CS) is the only predictor set that restricts the variety of all nonendemic and endemic gymnosperm species in China [18]. A substantial and significant negative association was discovered in this study between CS variables (temperature seasonality, TAR, and MDT) and gymnosperm richness (Figure 8). Previous studies have found a substantial negative association between climate seasonality and species richness [17,18,67,74,75,76], which is similar to our findings. At various elevations, species diversity is limited by topographic variability and climate. Furthermore, climatic seasonality (CS) is the only indicator of climatic variables limiting gymnosperm species richness in China, demonstrating that seasonal temperature fluctuation has a substantial impact on species richness [18,74]. Plants are stressed when altitude increases due to greater seasonal climate change, which may affect the root system’s capacity to absorb water, resulting in drought stress. According to Körner [35], both temperature and precipitation decrease with elevation, limiting the quantity of water and energy available to plants. Temperatures on the plateau vary substantially from southeast to northwest (Figure 9c), with the northern areas seeing lower maximum temperatures than the southern areas. This extreme temperature may be the explanation for the limited spread of gymnosperm species in the north QTP. The challenge for gymnosperms to acclimate to the extreme temperature and their inability to shift from southern areas to northern areas corroborate the niche conservatism idea in the tropical region [77].

### 3.3. Gymnosperms’ Conservational Aspects along Altitudinal Gradients

The high percentage of endangered species highlights the region’s importance for the protection of a diverse range of species in China and throughout the world [78]. Almost one third (69 species) of China’s total 195 gymnosperms are threatened, according to Xie et al. [10], while 29 species of gymnosperms on the Qinghai–Tibet plateau are designated as vulnerable, near threatened, or endangered (Table 1 and Figure 3S). Wu [79] found that 41% of species will be at risk of extinction due to climate change after examining the uncertainty and risk of habitat loss for 109 gymnosperms in China. For example, *Ginkgo biloba*, *Picea likiangensis var. hirtella*, *Amentotaxus assamica*, and *Taxus wallichiana* are all endangered species found in high-elevation zones. According to Xie et al. [48], endangered gymnosperm hotspots are usually situated on mountainous terrain and will diminish and relocate northward due to climate change. Similar to our findings, climatic seasonality factors (e.g., temperature seasonality and TAR) are the key variables determining the distribution of the most threatened gymnosperms [10,48]. Because of the effect of the cold environment, the growing period of some gymnosperm species (e.g., *Thuja sutchuenensis* and *Abies ziyuanensis*) was prolonged and reproductive capability declined [80], threatening and narrowing the distribution of these unique conifers in China [48]. Furthermore, the fluctuation in temperature caused by temperature seasonality may disrupt plant development and growth cycles, resulting in plant mortality [81,82]. As a result of climate change, species at the highest elevations are shifting up-slope, resulting in greater fragmentation and range loss, and maybe even “mountain peak extinction” [83,84,85,86]. This impact is most strong in species with limited distribution ranges (e.g., *Abies* spp.) that are restricted to mountain peaks.

Ecologists can contribute at every stage of a conservation effort, from defining the problem and team members to establishing goals and objectives, developing conservation techniques, and assessing progress. Conservation projects are more likely to succeed if they are socially and environmentally aligned with the local context [87,88]. Researchers and practitioners in gymnosperm conservation should collaborate to build a research agenda that addresses current knowledge gaps on successful conservation measures and ways to engage more effectively. We suggest that climate change conservation plans include not just regions critical for preserving present species diversity, but also areas critical for future adaptation. The most recommended measures include increasing the size of nature reserves to cover more high priority regions and climatically suitable habitats of vulnerable species and carrying out ex situ conservation to maintain and establish new populations in suitable habitats [89].

The distribution of low-altitude gymnosperm species (below 1000 m) may be particularly vulnerable to climate change effects [48] and land use variation due to increased human disturbance [90]. According to Yu et al. [91], 48% of species’ geographic ranges will shrink, with species loss predominating at lower altitudes. *Cephalotaxus*, for example, occurs at low elevations and is mostly impacted by habitat degradation induced by agricultural development and logging [92]. Endangered gymnosperms are crucial indicators of natural ecosystem health due to global climate change, and different and diverse conservation methods should be implemented for threatened species with varied regional distribution patterns to mitigate the implications of global climate change.

## 4. Materials and Methods

### 4.1. Description of Study Area

The Qinghai–Xizang (Tibet) Plateau (QTP) is situated in southwest China (25°~40° N and 75°~104° E) and covers an area of 2.6 × 10^6^ km^2^, as shown in Figure 9. The west side of the QTP is higher in elevation than the center and eastern sections [93]. The average annual temperature ranges from −15 °C to 10 °C, with 392–764 mm of precipitation falling on average every year. QTP is the third pole in the world, after the Arctic and Antarctic, and has a rich diversity of species, ecological types, and climates [94]. The variety of habitats also demonstrates how species richness varies dramatically across the plateau [95]. The QTP is home to about 9000 plant species, with over 18% of them being indigenous and extremely sensitive and susceptible to climate change [96]. The high elevation and complicated plateau terrain have a considerable thermal impact on the air temperature, circulation, and climatic patterns on the northern slope [97].

The Qinghai–Tibet Plateau was chosen as the research site because the Himalayan, Hengduan, Qilian, and Kunlun Mountains in China are home to a broad range of gymnosperm species. We collected data on the elevation and distribution range of gymnosperm species using “The vascular plant and their ecogeographical distribution of the Qinghai-Tibetan Plateau” [98], and we used the taxonomy and nomenclature from the “Color Atlas of Vascular Plants in Qilian Mountain” for gymnosperm species [99]. The databases and checklists stated above are based on distribution data of plant species in the QTP that several local and national teams have collected over the preceding 60 years [99].

To improve the databases’ accuracy, we also checked the plant record “www.plantplus.cn” (accessed on 25 July 2023), provincial records, published works, and regional floral diversity. Almost all of the counties on the QTP have undergone a comprehensive investigation, according to the new examination of the collection integrity of plant specimens [95]. On the QTP, all gymnosperm species were distributed between 600 m and 5300 m based on our preliminary research. The elevation gradient of gymnosperm distribution was used to form 53 elevation bands, each 100 m wide. A taxon presence was defined as being present every 100 m along an elevational gradient between the higher and lower elevations. A taxon with an altitude range of 840 to 1250 m, for example, can be present in bands of 900, 1000, 1100, 1200, and 1300 m. This results in anticipated gamma diversity, which quantifies overall richness along elevations [100].

### 4.2. Species Abundance and Distribution

Species abundance or richness, which is the independent variable or response variable in this research study, is determined as the abundance of gymnosperms placed in each band as assessed using interpolated techniques. The species abundance was approximately estimated based on the species distribution range along elevation gradients. This approach assumed that species might be found everywhere between their highest and lowest elevations. The under-sampling issue can be solved using the interpolation method, as well. Furthermore, due to the mountain’s hump-shaped structure, the surface area will be greater at the base than at the peak [35]. Each band will have a distinct surface area, and the richness of species will vary as land area varies over height. Furthermore, the surface area included inside each band is a substitute factor for the overall proportions of a gene pool and has significant effects on species richness [101]. The Shuttle Radar Topography Mission (SRTM; https://www2.jpl.nasa.gov/srtm/ [accessed on 12 June 2023]) provided the regional data on elevation and area.

The elevation range (upper and lower elevation) and the regional distribution at the city or county level are the two domains that characterize the geographic range of each species. There are 139 counties in QTP, all of which have various types of topography. The diversity maps of each taxon were then transformed into grid maps to lessen the possible impact of surface area on the assessment of species abundance. The plateau was split into 2,639,621 grids (1 km × 1 km) to accomplish this. To determine the elevation and specific point for each grid, we then integrated this grid map with a Chinese administrative map and a DEM (at 1 arc 2nd resolution). A taxon was found in the grid after evaluating both the vertical and horizontal distribution of a species. Because some elevation grids cover many counties, the species’ distribution range can be expanded based on horizontal distribution.

### 4.3. Climatic Variables

The variation in environmental variables is directly or indirectly determined by elevation, and these variables have a direct effect on plant growth and development [36]. We used eighteen environmental parameters to examine the gymnosperms’ richness patterns across elevation gradients. Some significant variables were projected onto 1 km grids on the basis of elevational data from the “Shuttle-Radar Topography Mission” (SRTM), using a spline-based technique (ANUSPLIN ver. (4.4), Xu and Hutchinson [102]), which took into account the effect of altitude on climates. Based on the purpose of the investigations, we divided these factors into 3 key variable sets, namely (i) climatic seasonality (CS), (ii) energy–water (EW), and (iii) physical tolerance (PT). These variable sets were also used in previous studies to evaluate the pattern of species richness of gymnosperms along elevation gradients [17,18].

Climatic data from 2152 meteorological stations in China were collected from long-term records from 1981 to 2010 “http://data.cma.cn” (accessed on 5 May 2023). According to previous research, the “accumulated growing degree days above 5 °C” (GDD_5_), “accumulated growing degree days above 0 °C” (GDD_0_), “sum of growing season precipitation” (GP), “annual precipitation” (AP), “annual mean temperature” (MAT), “percentage of solar radiation” (SSP), “moisture index” (MI), and “drought index” (DI) are the substantial proxy variables indicating the water and energy sources [40,44]. Equations (1) and (2) were used to compute the MI and DI, respectively.
(1)MI=MAP/PET
(2)DI=1−AET/PET
where “PET” refers to the “annual potential evapotranspiration” and “AET” refers the “annual actual evapotranspiration” calculated using the “Penman Monteith” method [103].

As surrogate variables for physical tolerance (PT), we used the “mean minimum temperature” (Tmin), “mean temperature of the coldest month” (MTCM), “mean maximum temperature” (Tmax), “mean temperature of the warmest month” (MTWM), “precipitation of the driest month” (PDM), and “precipitation of the wettest month” (PWM). The use of comparable factors in estimating plant species richness has been confirmed by Jiang et al. [104] and Gao and Liu [40]. The data from the meteorological stations were used to compute all of the proxy variables for physical tolerance.

Finally, “precipitation seasonality” (PS, %), “temperature seasonality” (TS, °C), “annual temperature range” (TAR, °C), and “annual mean diurnal range” (MDR, °C) were climatic seasonality proxy variables. In previous research, these variables were also utilized to describe climate seasonality [44].

### 4.4. Statistical Analysis and Correlation

A ternary plot was used to determine the combined and individual percentages of each predictor set in order to access the overall effect of each predictor and their aggregate contributions to more effectively define the richness patterns of gymnosperm species. The altitudinal gradient was used as an explanatory variable, while species richness was chosen as a response variable.

The Pearson correlation coefficient was used to determine a significant relationship among species richness, elevation, and environmental variables. Further information on the primary factors affecting species diversity may be obtained by examining the richness pattern at the two ends of the whole gradient. We used PCA to determine the differences between two sub-gradients in order to find a substantial relationship between gymnosperm richness and environmental variables. The elevation gradient was used as an explanatory variable, while the environmental variables and the richness of gymnosperm species were used as response variables. Autocorrelation was used to demonstrate the relationship between gymnosperm species richness and climatic parameters with altitude in PAST 4.12b. All graphical data analysis was performed using R software 4.2.3, Microsoft Excel 365 (Microsoft, Redmond, WA, USA), and PAST 4.12b [105].

We used a “generalized linear model” (GLM) to find the correlation between gymnosperm richness and environmental variables along the elevation gradients. Previous research used the “GLM” model to establish correlations between climatic conditions and species richness [16,33]. The regression models were also examined using an identity link function, which employs a poison distribution error. Each proxy variable was examined separately as well as with respect to null models. It was feasible to assess whether the fitted statistics were adequate by using normal probability graphs. We may analyze the relative impact of climatic seasonality (CS), energy–water (EW), and physical tolerance (PT) in determining gymnosperm richness patterns along the steepest elevational gradients on QTP using the “GLM” model and variation partitioning.

## 5. Conclusions

This study explored the vertical and horizontal distribution of gymnosperms on the QTP for the first time and revealed the overall effect of environmental variables in explaining the distribution patterns of species. According to the findings, gymnosperms formed a mid-elevation curve with a diminishing horizontal distribution pattern from southeast (SE) to northwest (NW). Our results showed that the relative significance of physical tolerance (PT) and energy–water (EW) parameters determined the mid-elevation curve of gymnosperm richness. Climatic seasonality (CS) is an influential factor that has resulted in a decrease in gymnosperm species richness at higher elevations on QTP. The substantial relationship between environmental factors and gymnosperm richness underlined the significance of a multi-gradient study for assessing species distribution patterns along altitudinal gradients. The global climate change event may impact the montane biota, causing a shift in range distribution and a decrease in species richness. Despite the fact that threatened gymnosperm species account for a relatively large proportion, immediate effort is essential to conserve vulnerable species with various regional distribution patterns in order to counterbalance the consequences of global climate change. More investigation is required to develop conservation strategies for the development of protected areas for ecologists and policymakers. Furthermore, future research should concentrate on predicting threatened gymnosperm richness patterns as a result of climate change, as well as how they adapt to seasonal climate fluctuations. Nevertheless, because these assumptions are dependent on predicted species and climatic data, they must be validated by actual sampling from established sample plots with observable climatic parameters.

## Figures and Tables

**Figure 1 plants-12-04066-f001:**
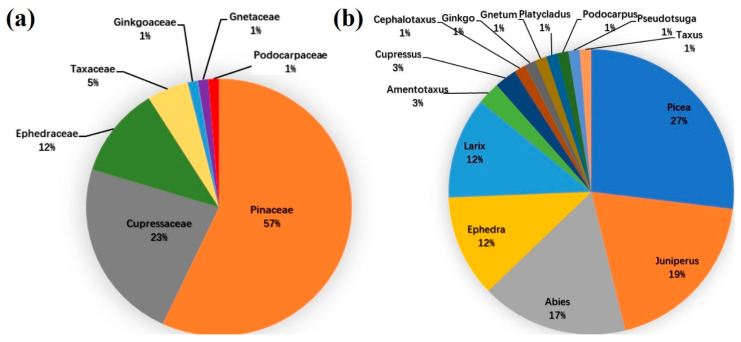
Percentages of (**a**) families and (**b**) genera of gymnosperm species in the Qinghai–Tibet Plateau.

**Figure 2 plants-12-04066-f002:**
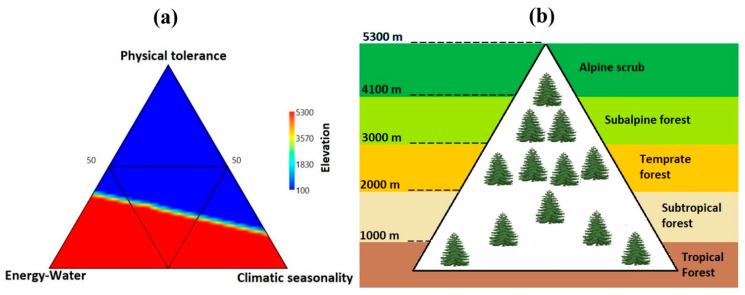
(**a**) Ternary plot shows the relative contributions of climatic seasonality (CS), energy–water (EW) and physical tolerance (PT) to elevation in the Qinghai–Tibet Plateau. (**b**) Diagram depicting habitats in the Qinghai–Tibet Plateau. The alpine region comprises the alpine and subalpine zones. The subalpine zones have the greatest diversity of gymnosperm species, at an elevation of 3200 m.

**Figure 3 plants-12-04066-f003:**
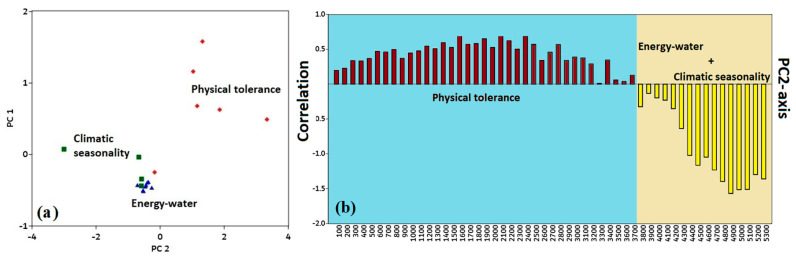
(**a**) Environmental variables in the principal component analysis plotted in relation to elevation gradients in the Qinghai–Tibet Plateau, including climatic seasonality (CS), energy–water (EW), and physical tolerances (PT). (**b**) Loadings of variables in PCA show the correlation with the PC2-axis.

**Figure 4 plants-12-04066-f004:**
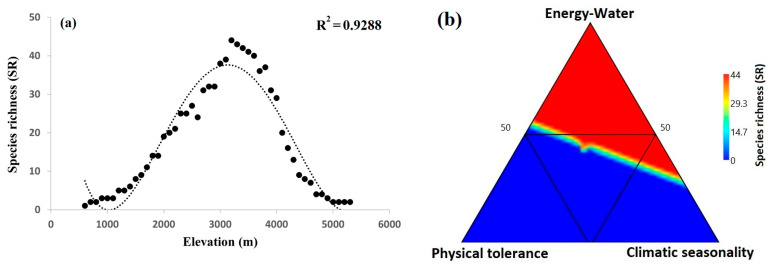
(**a**) Gymnosperm species richness in the Qinghai–Tibet plateau (black circles) with the best fit of the polynomial curve (dotted line). The explanatory power of the regression model is represented by R2 value (*p* < 0.001). (**b**) Ternary plot shows the relative contributions of climatic seasonality (CS), energy–water (EW) and physical tolerances (PT) to patterns of gymnosperm species distribution along the elevation gradients in the Qinghai–Tibet Plateau.

**Figure 5 plants-12-04066-f005:**
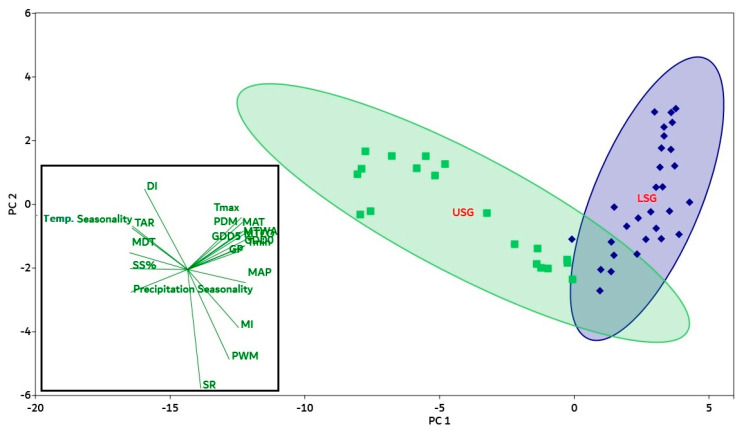
Plot of environmental variables in the PCA conducted with GDD5 (annual growing degree days above 5 °C), GDD0 (annual growing degree days above 0 °C), GP (growing season precipitation), AP (annual precipitation), MAT (annual mean temperature), SSP (solar radiation %), MI (moisture index), DI (drought index), Tmin (minimum temperature), MTCM (minimum temperature of the coldest month), Tmax (maximum temperature), MTWM (maximum temperature of the warmest month), PDM (precipitation of the driest month), PWM (precipitation of the wettest month), PS (precipitation seasonality), TS (temperature seasonality), TAR (annual temperature range), and MDR (annual mean diurnal range) in lower sub-gradients (LSG, purple color) and upper sub-gradients (USG, green color) in the Qinghai–Tibet Plateau. Based on the mid-elevation curve at 3200 m, the whole gradient was divided into two groups of 100–3200 m (LSG) and 3300–5300 m (USG), and both were analyzed simultaneously in the principal component analysis. Loadings of environmental variables in PCA show the correlation with PC1 and PC2. The length of the green line represents the total contribution of climatic factors to the analysis. The direction of the green line illustrates the association of environmental variables with each axis (vector lines parallel to an axis are significantly connected with that axis). Correlations between climatic factors are shown by the angles between vector lines.

**Figure 6 plants-12-04066-f006:**
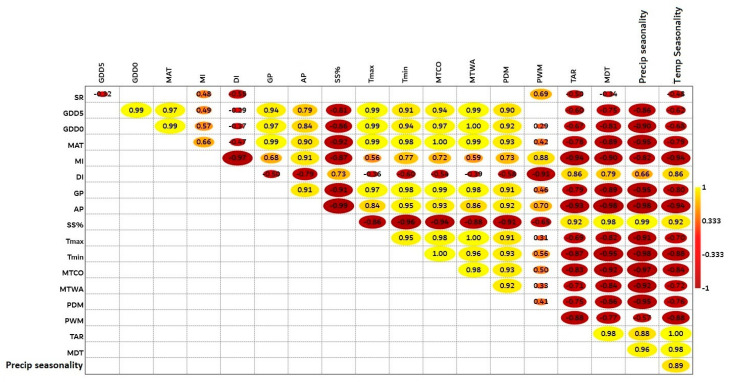
Pearson correlation coefficient shows the correlation between climatic conditions and species richness. The filled box indicates a significant association (*p* < 0.05), while the empty box indicates a non-significant correlation. The red circles indicate a negative significant correlation (*p* < 0.05), whereas the yellow circles indicate a positive significant correlation (*p* < 0.05).

**Figure 7 plants-12-04066-f007:**
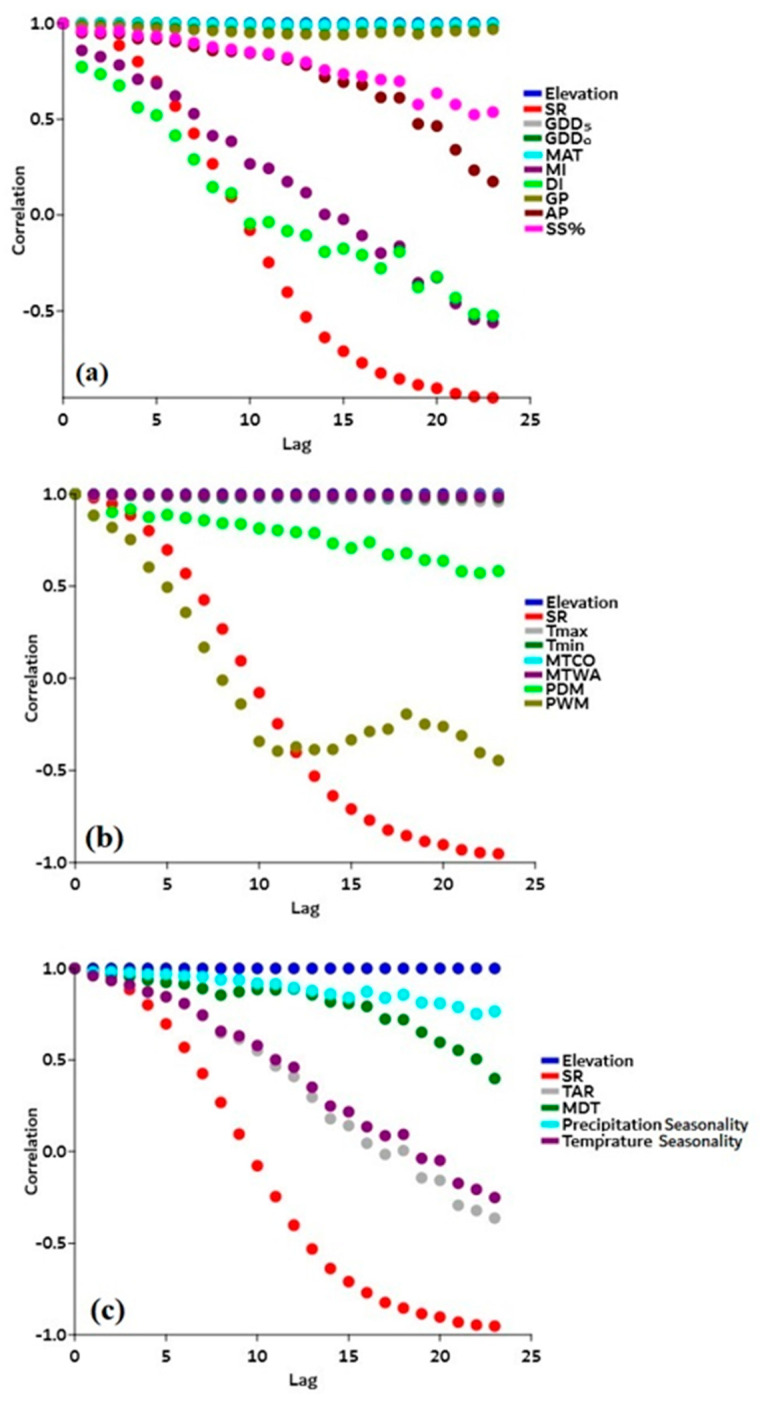
Spatial autocorrelation graphs show the significant association between predictor variables (dependent variables), species richness (SR), and elevation (independent variables). (**a**) Energy–water variables (i.e., GDD_5_ (growing degree days of daily temperature > 5 °C, gray dots), GDD_0_ (growing degree days of daily temperature > 0 °C, dark green dots), MAT (mean annual temperature, aqua dots), GP (growing degree days, golden dots), AP (annual precipitation, maroon dots), SSP (sunshine %, light green dots), MI (moisture index, purple dots), and DI (drought index, violet dots)) show significant association with species richness (SR, red circles) and elevation (blue circles). (**b**) Physical tolerance variables (i.e., minimum temperature (Tmin, gray dots), minimum temperature of the coldest month (MTCM, purple dots), maximum temperature (Tmax, dark green dots), maximum temperature of the warmest month (MTWM, aqua dots), precipitation of the driest month (PDM, light green dots), and precipitation of the wettest month (PWM, golden dots)) show significant association with species richness (SR, red circles) and elevation (blue dots), and (**c**) climate seasonality (CS) (i.e., the annual temperature range (TAR, gray dots), annual mean diurnal range (MDR, dark green dots), precipitation seasonality (PS, aqua dots), and temperature seasonality (TS, purple dots)) show significant association with species richness (SR, red dots) and elevation (blue dots).

**Figure 8 plants-12-04066-f008:**
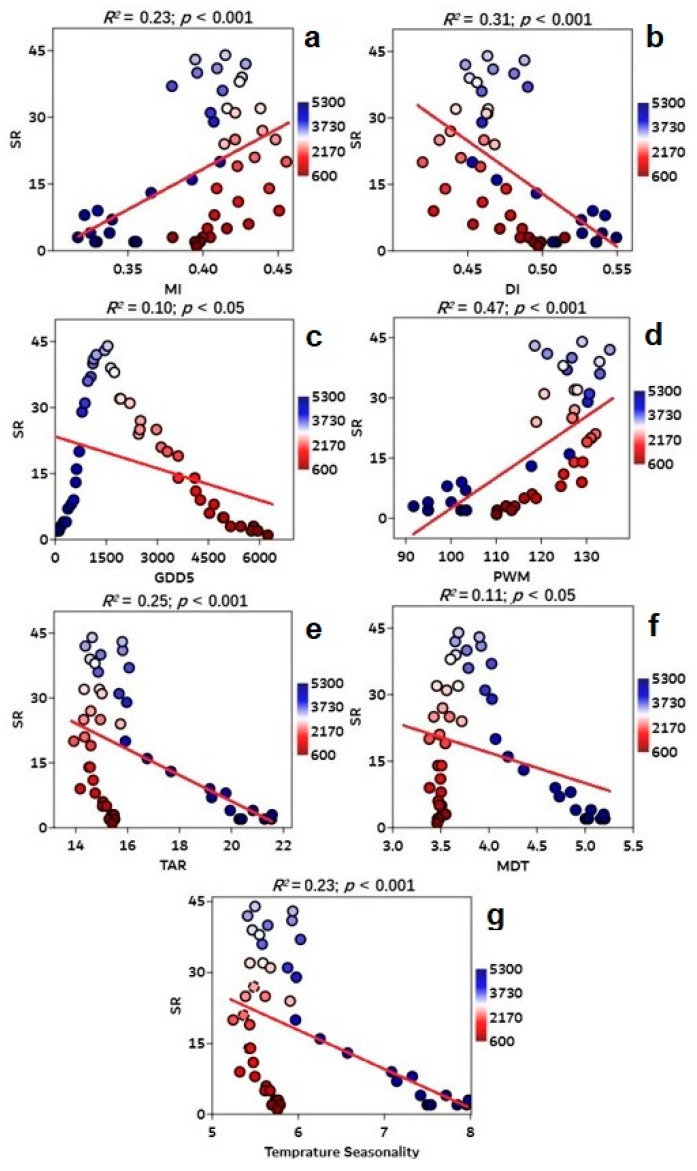
Scatter plots showing the significant relationship between species richness on *y*-axis and various climatic variables along the elevation gradients, i.e., (**a**) MI (moisture index), (**b**) DI (drought index), (**c**) GDD5 (annual growing degree days above 5 °C), (**d**) PWM (precipitation of the wettest month), (**e**) TAR (annual temperature range), (**f**) MDR (annual mean diurnal range), and (**g**) temperature seasonality on *x*-axis. Regression models describe the significant link between species richness and climate variables by fitting a red line to the data. The significant R^2^ value represents the regression model’s explanatory power. The scale color represents the length of the elevation gradients (100–5300 m a.s.l.) in the Qinghai–Tibet Plateau, China.

**Figure 9 plants-12-04066-f009:**
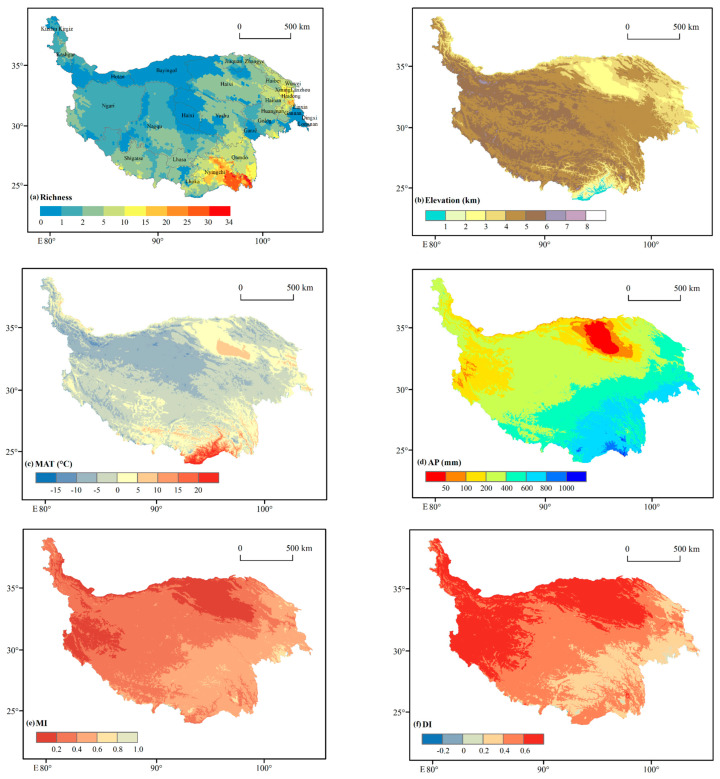
The digital elevation model: (**a**) gymnosperms richness, (**b**) altitude, (**c**) mean annual temperature (MAT), (**d**) annual precipitation (AP), (**e**) moisture index (MI), and (**f**) drought index (DI) in the Qinghai–Tibet Plateau.

**Table 1 plants-12-04066-t001:** Native distribution range and conservation status of gymnosperm species. The mean elevation is the arithmetic mean of the minimum and maximum elevations.

Family	Species	IUCN Status	Mean Elevation (m)	Haibei TAP	Huangnan TAP	Hainan TAP	Guoluo TAP	Yushu TAP	Haixi Mongolian TAP	Xining City	Haidong City	Nyingchi City	Qamdo City	Shigatse City	Lhoka City	Lhasa City	Nagqu City	Ngari City	* Global Distribution Range
Cupressaceae	*Cupressus gigantea*	VU	3200									√							±±
	*Cupressus torulosa*	LC	2250									√							±±±
	*Juniperus formosana*	LC	2600	√	√						√	√	√						±±±
	*Juniperus recurva*	LC	2950		√							√		√	√				±±±
	*Juniperus rigida*	LC	2000							√									±±±
	*Juniperus communis var. saxatilis*	NT	3850											√					±±±±±
	*Platycladus orientalis*	NT	2000		√					√	√	√							±±±
	*Juniperus chinensis*	LC	1850	√					√	√		√	√	√		√			±±±±
	*Juniperus convallium*	LC	3650				√	√		√			√						±±
	*Juniperus komarovii*	LC	3750				√												±±
	*Juniperus microsperma*	DD	3725					√				√							±±
	*Juniperus pingii var. wilsonii*	NT	4350					√				√	√	√	√	√			±±
	*Juniperus przewalskii*	LC	3275	√	√	√	√			√	√								±±
	*Juniperus saltuaria*	LC	3700								√	√	√						±±
	*Juniperus squamata*	NT	3170							√	√	√		√	√				±±±±
	*Juniperus tibetica*	VU	3850		√		√	√				√	√	√	√	√	√		±±
	*Juniperus sabina*	LC	3200	√		√			√		√							√	±±±±±
	*Juniperus indica*	LC	4200										√	√					±±±±
Ephedraceae	*Ephedra equisetina*	NE	2400			√			√	√	√								±±±±
	*Ephedra gerardiana*	VU	4350	√				√	√	√			√	√	√	√	√	√	±±±±
	*Ephedra intermedia*	LC	2975		√	√		√	√	√	√	√	√		√			√	±±±±
	*Ephedra likiangensis*	LC	3200										√						±±
	*Ephedra saxatilis*	LC	3900									√		√	√	√			±±±
	*Ephedra minuta*	LC	3500	√	√		√	√	√	√	√								±±±
	*Ephedra monosperma*	LC	4000			√	√	√	√		√	√	√			√	√	√	±±±±
	*Ephedra przewalskii*	LC	3000			√			√										±±±±
	*Ephedra sinica*	LC	2850		√	√			√		√								±±±±
Ginkgoaceae	*Ginkgo biloba*	EN	1975							√	√								±±
Gnetaceae	*Gnetum pendulum*	LC	800									√							±±
Pinaceae	*Abies chayuensis*	LC	1940									√							±±
	*Abies delavayi var. delavayi*	LC	1600									√							±±±
	*Abies delavayi var. motuoensis*	LC	3150									√							±
	*Abies densa*	LC	3230											√					±±±
	*Abies ernestii*	VU	3300				√						√						±±
	*Abies ernestii var. salouenensis*	LC	2900									√							±±
	*Abies fargesii*	LC	2850								√								±±
	*Abies fargesii var. faxoniana*	LC	3400				√				√								±±
	*Abies forrestii*	LC	3800									√							±±
	*Abies georgei*	LC	3875									√							±±
	*Abies georgei var. smithii*	LC	3450									√							±±
	*Abies spectabilis*	NT	3300											√					±±±
	*Abies squamata*	VU	3600				√	√					√						±±
	*Larix gmelinii*	LC	2500	√		√				√	√								±±±±
	*Larix griffithii*	LC	3400									√		√	√		√		±±±
	*Larix himalaica*	NT	3200											√					±±±
	*Larix kaempferi*	LC	1350		√					√									±±±
	*Larix olgensis*	NT	1350							√									±±±
	*Larix potaninii*	LC	3750				√				√	√	√						±±±
	*Larix potaninii var. australis*	LC	3950									√	√						±±
	*Larix gmelinii var. principis-rupprechtii*	LC	2500	√		√				√	√								±±
	*Larix speciosa*	NT	3500									√							±±
	*Picea brachytyla*	VU	3250		√		√					√	√		√				±±
	*Picea brachytyla var. complanata*	VU	3250									√	√		√				±±
	*Picea crassifolia*	LC	3100	√	√	√	√		√	√	√								±±
	*Picea likiangensis var. rubescens*	VU	4050				√	√				√	√				√		±±
	*Picea likiangensis var. hirtella*	EN	1900										√						±±
	*Picea likiangensis var. linzhiensis*	VU	3400									√			√				±±
	*Picea purpurea*	NT	3340		√		√	√		√	√								±±
	*Picea asperata*	VU	3450				√												±±
	*Picea schrenkiana*	LC	2350									√	√						±±±
	*Picea smithiana*	LC	2750											√					±±±
	*Picea spinulosa*	LC	3250											√					±±
	*Picea wilsonii*	LC	2700		√						√								±±
	*Pinus armandii*	LC	2600								√	√	√						±±±
	*Pinus bhutanica*	LC	1700									√							±±
	*Pinus densata*	LC	2850									√	√		√				±±
	*Pinus gerardiana*	NT	1350															√	±±±
	*Pinus wallichiana*	NT	2250									√		√	√				±±±
	*Pinus roxburghii*	LC	2200											√					±±±
	*Pinus sylvestris var. mongolica*	VU	1400							√	√								±±±
	*Pinus tabuliformis*	LC	2400	√	√	√				√	√								±±±
	*Pinus yunnanensis*	LC	2000									√							±±
	*Pseudotsuga forrestii*	VU	3000									√							±±±
	*Tsuga dumosa*	LC	2600									√		√	√				±±±
Podocarpaceae	*Podocarpus neriifolius*	LC	1075									√							±±±±
Taxaceae	*Cephalotaxus mannii*	VU	1150									√				√			±±±±
	*Amentotaxus argotaenia*	NT	800									√							±±±
	*Amentotaxus assamica*	EN	850									√							±±
	*Taxus wallichiana*	EN	2700									√		√					±±±±

* The global distribution range of gymnosperm species is given in Appendix A. It is represented by “±”, and the more “±”, the wider the distribution range, and five “±” indicate distribution throughout the Eurasian continent. “√” sign shows the presence of a species in a specific region.

## Data Availability

All the data are supplied in tables and figures in the publication or as Appendix A, and any further questions should be referred to the relevant authors.

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
