# Peer review of "Distribution Patterns of Gymnosperm Species along Elevations on the Qinghai–Tibet Plateau: Effects of Climatic Seasonality, Energy–Water, and Physical Tolerance Variables"

_plants, 2023, doi:10.3390/plants12234066_

Round 1
Reviewer 1 Report
Comments and Suggestions for Authors
Dear Authors,
Please receive my appreciation for your hard and accurate work. As a result, I don't have suggestions regarding the structure, but some corrections, as follows:
1. In more than one place (rows 210-211, 459-460, 497 and others), you mention the name of all the cited authors, then giving the citation number - it doesn't look good.
2. In rows 233-234 you even mention "(Hammer et al., 2001)", eventhough the title isn't mentioned in References.
3. In rows 180 you wrote "girds", instead of "grids".
4. The title of Figure 1 isn't correct - please check and fix that.
5. Figure 8 is hard to understand, since not all the dots are visible, at least in (a) and (b).
6. Figure 9 includes several graphs, but without being explained by letters, as the others are.
7. PLEASE CHECK AGAIN THE TEMPLATE FOR "PLANTS"! The "Materials and Method" section should be in Point 4.
Thank you for considering my opinion!
Best regards,
Comments on the Quality of English LanguageDear Authors,
The English language is fine, but please check again the entire manuscipt, in order to find and correct the wrong words, missing or adding spaces, subscript/superscript and so on.
Thank you in advance!
Best regards,
Author Response
Dear Reviewer,
Response: Thank you very much for your comments and suggestions. The comments and suggestions are valuable and very helpful for revising and improving our manuscript. We have revised this article according to the given comments and have made changes as you directed us (marked red).

Reviewer 2 Report
Comments and Suggestions for Authors
The introduction incuded relevant references, all the cited references are relevant to the research.
The research design is appropriate and the methodes are adequately described.
The results are well and clear presented and the conclusions are supported by the results.
Author Response
Dear Reviewer,
Thank you very much for your comments and suggestions. The comments and suggestions are valuable and very helpful for revising and improving our manuscript. We have revised this article according to the given comments and have made changes as you directed us (marked red).
Reviewer 3 Report
Comments and Suggestions for Authors
Dear Authors,
I have finished my review on this manuscript. Great job! My only concern is the untypical length of the paper which could benefit from a better synthesis. In rest, I have no major objections.
BR,
Rev.
Comments on the Quality of English LanguageLanguage is fine.
Author Response

(The authors gave the same response as above.)

Reviewer 4 Report
Comments and Suggestions for Authors
Major Comments:
- The introduction provides good background on gymnosperm distributions and climate change impacts on the Tibetan Plateau, but needs a clearer overview of the gap this study aims to fill and its novel contributions. The authors should add a thesis statement previewing the major goals, approaches, and findings of their research.
- In the methods, more justification is needed for the specific environmental variables chosen and their grouping into the CS, EW, and PT predictor sets. Are these standard groupings in literature? Why were these particular factors selected over others?
- The discussion could be strengthened by comparing results to similar elevational diversity studies on gymnosperms or other taxa in nearby regions like the Himalayas. Are the distribution patterns and climate correlations found here consistent with prior work?
- The conservation implications discussed are interesting but quite brief. The authors could expand on specific strategies needed to protect threatened gymnosperm species based on their elevational ranges and climate limitations found in this study.
- More explanation of the limitations of this work is needed in the conclusion - e.g., possible biases in underlying distribution data, limitations of Granger causality, need for phylogenetic and biotic analyses to understand mechanisms.
Additional weaknesses include haphazard overuse of abbreviations, figures that are blurry and lack clarity, and persistent grammar issues throughout. The writing requires substantial refinement for clarity and flow. References are inconsistently formatted. Overall, while the analysis itself seems sound, the framing, interpretation, and presentation of the work falls well short of expectations.
Minor Comments:
- Avoid overuse of abbreviations (e.g., QTP, SE, NW) in the abstract where possible. Write out full terms.
- Carefully proofread the manuscript to fix minor typos, grammar issues, and awkward wording throughout.
- Make sure all abbreviations are defined before first use.
- Unclear sentence: "According to native distribution of gymnosperm species, Juniperus communis var. saxatilis and Juniperus sabina are widely spread over the Eurasian continent." Recommend revising for clarity.
- Redundant phrasing: "Pearson correlation coefficient data revealed that..." Could shorten to "Pearson correlation coefficients showed..."
- Wordy sentence: "As an explanatory variable, the elevation gradient was utilized, whereas gymnosperm species richness and environmental variables were used as response variables." Simplify wording.
- Unclear connection: "Figure 4b shows a positive association between the PC2 component and the physical tolerance (PT) variable..." Link the figures back to the results more directly.
- Repetitive wording: "Previous research has been used the “GLM” model is used to create correlations between species richness and climatic factors"
- Unclear phrasing: "Topographic variability and Climate limit the species diversity at various elevations." Rephrase for clarity.
- Redundant sentence: "According to evidence, as elevation increases, temperatures and precipitation decrease, limiting the amount of water and energy available to plants."
- Wordy phrasing: "The distribution of the highest gymnosperm diversity in the southeastern QTP might be related to differences in climatic conditions and elevation ranges." Simplify.
- Unclear connection: "Our findings imply that the gymnosperms richness patterns on QTP was determined by energy-water (EW) dynamics." Link back to specific results.
In summary, major revisions are required to address the critical issues outlined above related to the introduction, methods justification, discussion of key contexts and limitations, presentation quality, and writing. However, the analysis provides valuable insights that could benefit the literature if framed and contextualized thoroughly. I look forward to seeing the authors comprehensively improve the manuscript through major revision. Please feel free to contact me with any questions.
Comments on the Quality of English Language- Overuse of wordy, ambiguous, and redundant phrasing throughout, such as "Pearson correlation coefficient data revealed that..." or "According to native distribution of gymnosperm species..."
- Multiple grammatical errors, including subject-verb disagreement, punctuation issues, and sentence fragments. The writing lacks polish.
- Frequent overly complex or unclear sentences, like "Topographic variability and Climate limit the species diversity at various elevations."
- Inconsistent style for abbreviations and scientific terms - sometimes abbreviated, sometimes written out.
- Missing or unclear connections between ideas, figures, and results. The flow of logic is not always smooth.
- Various typos, spelling inconsistencies, and formatting issues.
The writing style comes across as repetitive, convoluted, and careless overall. The authors need to thoroughly proofread and edit the entire manuscript to improve clarity, precision, and grammatical correctness. Sentences should be simplified and smoothed out. Attention should be paid to using clear language that concisely connects ideas and supports comprehension. Significant refinement of the English language and presentation is required before this manuscript would be suitable for publication. Please feel free to contact me for any clarification or examples of suggested revisions.
Author Response

(The authors gave the same response as above.)

Reviewer 5 Report
Comments and Suggestions for Authors
This manuscript is interesting but needs to be improved. The critical comments on the manuscript are given as following.
1. Line 14, “the Qinghai-Xizang (Tibet) Plateau…” should be “the Qinghai-Xizang (Tibet) Plateau (QTP)…”
2. In the Introduction, the authors should highlight the importance and novelty of this study.
3. The assumption of this study would be addressed.
4. Line 281, “Figure S4 and S5” should be “Figures S4 and S5”.
5. Line 307, “r2=0.92” would be “r2=0.929”.
6. Line 356, “Figure 8a and 8b” should be “Figures 8a and 8b”.
7. In Figure 9, the R-squared between SR and other parameters. More explanations for the low R-squared are needed.
8. Line 444, “…..along elevational gradient serves…” should be “…..along elevational gradient serve…”
9. Line 464, “Figure 1c and 1d” should be “Figures 1c and 1d”.
10. Lines 476-477, “According to S Wang, F Liu, Q Zhou, 476 Q Chen, B Niu and X Xia [80],…” should be revised as “According to Wang et al. [80],…”
11. Lines 482-483, “According to Fei et al. [83] research,”
12. The authors should check lines 490-491, line 497, lines 532-434, lines 439-440, and lines 561-562. They have same problems like previous comment.
13. Line 535, “Wu (2020)” should use number.
Comments on the Quality of English LanguageModerate editing of English language required.
Author Response

(The authors gave the same response as above.)

Round 2
Reviewer 4 Report
Comments and Suggestions for Authors
This paper shows promise and can now be considered publishable after the revision.
Reviewer 5 Report
Comments and Suggestions for Authors
The revised manuscript is acceptable. I recommend that it can be accepted for publication.